# Glioma Stem Cells Are Sensitized to BCL-2 Family Inhibition by Compromising Histone Deacetylases

**DOI:** 10.3390/ijms241813688

**Published:** 2023-09-05

**Authors:** Aran Merati, Spandana Kotian, Alexus Acton, William Placzek, Erin Smithberger, Abigail K. Shelton, C. Ryan Miller, Josh L. Stern

**Affiliations:** 1Department of Biochemistry and Molecular Genetics, University of Alabama at Birmingham, Birmingham, AL 35294, USA; 2O’Neal Comprehensive Cancer Center, Birmingham, AL 35294, USA; 3Department of Pathology, Division of Neuropathology, University of Alabama at Birmingham, Birmingham, AL 35294, USA

**Keywords:** glioblastoma, apoptosis, HDAC inhibitor, BIM, BMF, BCL-2 family, MEK1/2, hyaluronic acid

## Abstract

Glioblastoma (GBM) remains an incurable disease with an extremely high five-year recurrence rate. We studied apoptosis in glioma stem cells (GSCs) in response to HDAC inhibition (HDACi) combined with MEK1/2 inhibition (MEKi) or BCL-2 family inhibitors. MEKi effectively combined with HDACi to suppress growth, induce cell cycle defects, and apoptosis, as well as to rescue the expression of the pro-apoptotic BH3-only proteins BIM and BMF. A RNAseq analysis of GSCs revealed that HDACi repressed the pro-survival BCL-2 family genes MCL1 and BCL-XL. We therefore replaced MEKi with BCL-2 family inhibitors and observed enhanced apoptosis. Conversely, a ligand for the cancer stem cell receptor CD44 led to reductions in BMF, BIM, and apoptosis. Our data strongly support further testing of HDACi in combination with MEKi or BCL-2 family inhibitors in glioma.

## 1. Introduction

Glioblastoma (GBM) is the most frequent and aggressive primary brain malignancy and is classified as a grade IV isocitrate dehydrogenase (IDH) 1 or 2 wildtype glioma [1]. Despite involving the employment of a combination of surgery, radiotherapy, and chemotherapy, the standard of care results in a median survival of less than 16 months [2]. Genetic alterations in *EGFR*, *PTEN*, *TP53*, *ERBB2*, and *NF1* have been identified as pathogenic drivers in GBM, but precision therapies against these have not been successful so far in the clinic [3,4,5,6,7,8].

Gene expression analyses have shown that glioblastoma cells exist in diverse transcriptional states, which are classified as proneural, neural, classical, and mesenchymal [9,10,11,12,13,14]. GBM also harbors a subset of cancer stem cells, which are defined as cells with the capacity to initiate new tumors. This subset has been intensely investigated as a possible source of post-therapy recurrent tumors [15,16,17,18,19]. Of the subtypes, the mesenchymal class frequently displays particularly aggressive characteristics, including stem cell features, tumor initiating capacity in xenograft models, an association with tumor relapse, and reduced patient survival [10,13,20]. Thus, this glioma cell type remains a high priority for treatment. Prior studies have shown that successful cancer treatment will need to rely on targeting several oncogenic pathways using different approaches to prevent tumor cell resistance and recurrence [10,21].

One approach is to counter the mesenchymal and stem cell properties of GBM. Histone deacetylase inhibitors (HDACi) can block the cancer cell cycle [22] and alter the stem cell transcription program [15,23,24]. Thus, HDACi may have some capacity to transcriptionally reprogram GBM stem cells away from their pathogenic potential [23,25]. A second attractive target in GBM is the RAS/MEK/ERK pathway. The signaling of receptor tyrosine kinases (RTKs) such as epidermal growth factor (EGFR) and NF1 inactivation converges on MEK1 and MEK2, which activate their downstream effectors ERK1 and ERK2 [6,21,26,27]. A third attractive target is to sensitize GBM cells to apoptosis. Oncogenic RTK activation drives resistance to apoptosis in GBM in part by repressing the BH3-only pro-apoptotic protein BIM [28,29,30,31]. Activated MEK1/2 signaling triggers the proteasomal degradation of BIM, which inhibits apoptosis. The importance of this pathway in GBM cell survival has been shown in therapy-resistant GBMs that were resensitized to EGFR inhibition by targeting MEK1/2 [28]. BMF, another BH3-only pro-apoptotic protein, is also regulated by MEK1/2 [32]. In cancer cells, both BIM and BMF are frequently bound by BCL-2 anti-apoptotic proteins. This interaction inhibits the ability of BMF and BIM to initiate apoptosis. Peptides and small molecules that inhibit BCL-2 proteins have been developed and are able to rescue the ability of cancer cells to undergo apoptosis by releasing BH3-only proteins such as BIM and BMF [33].

Finally, microenvironmental factors also play major roles in regulating GBM cell survival [34,35,36,37]. One microenvironment factor with the potential to promote GBM progression and therapy resistance is the interaction between the stem cell marker CD44 and the glucosaminoglycan hyaluronic acid (HA), also known as. hyaluronan [38,39,40]. HA displays both autocrine and paracrine functions with the potential to support GBM cell survival [41,42,43].

In this study, we evaluated the effects of combining MEKi with HDACi in patient-derived xenograft stem cells (also known as glioma stem cells, GSCs). Our rationale was to use MEKi to rescue pro-apoptotic proteins to potentiate the effects of HDACi in dysregulating the cell cycle and altering the stem cell transcriptional program. To gain insight into the mechanisms driving this cell death, we analyzed changes in GSC transcriptome, key apoptotic proteins, and cell cycle factors. These analyses led to the discovery of a novel drug combination that enhanced GSC apoptosis.

## 2. Results

### 2.1. Inhibition of Histone Deacetylases Enhances MEKi-Induced Apoptosis in GBM

To examine the sensitivity of GBM to HDACi and MEKi, we used both patient-derived xenograft cells (PDX12, 39, herein referred to as GSC12, GSC39) and cell culture-adapted glioma lines (U87MG and LN229) (Appendix A). The PDX cells were grown under glioblastoma stem cell (GSC) serum-free conditions and the cell culture-adapted lines in serum-containing medium. For treatment, we selected clinically approved trametinib which inhibits MEK1/2 (MEKi), and vorinostat which inhibits class I, II, and IV HDACs (HDACi). MEKi treatment led to a moderate increase in cleaved caspase 3 (CC3) (Figure 1A,B and Appendix A) and a robust increase in the expression of pro-apoptotic BIM (Figure 1B and Appendix A). The GSCs displayed activated ERK (pERK) under these conditions, and this was suppressed by MEKi (Figure 1B). Vorinostat had a stronger effect than MEKi alone on CC3 activation, but the pro-apoptotic effect was most significantly enhanced with the combination treatment (Figure 1A,B and Appendix A). We observed similar effects with a range of HDAC inhibitors (Figure 1B). We conducted time-course studies of CC3 activation and analysis of phosphatidyl inositol membrane relocalization with annexin V, which demonstrated a significant percentage of cells undergoing apoptosis after 16–24 h (Appendix A). The apoptosis did not appear to be primarily caused by DNA damage, as blocking caspase activity prevented the accumulation of the DNA damage marker, pH2AX (Appendix A). The GBM apoptosis was associated with a reduction in cell growth and cell viability (Appendix A). The combination of the two drugs displayed synergistic effects on viability (combination index (CI) of <1) (Appendix A), which is consistent with our observations on apoptosis. Taken together, our results demonstrate the susceptibility of GBM in vitro to a combination of relatively low doses of these FDA-approved drugs.

### 2.2. MEKi+HDACi Rescues BIM and BMF Expression and Represses MCL1 and BCL-XL in GSCs

We next investigated two well-established triggers of apoptosis as potential mechanisms driving cell death in MEKi+HDACi treated GBM. These included cell cycle dysregulation and the rescue of pro-apoptotic function. We initially focused on apoptotic factors, as we previously observed that BIM protein expression was rescued by MEKi (Figure 1B). We performed a RNAseq analysis of GSC12 and U87 cells 24 h after treatment with MEKi, HDACi, or their combination. The mRNAs for several pro- and anti-apoptotic genes were significantly altered (GSC12 Figure 2A; U87 Appendix A), including BIM, BMF, BCL-XL, and MCL1. For pro-survival BCL-2 family proteins, the RNAseq analysis suggested that BCL-XL levels may be downregulated by the treatment. Subsequent immunoblot experiments validated the loss of BCL-XL protein in cell culture-adapted U87 and LN229 cells treated with MEKi+HDACi (Appendix A).

BIM mRNA and protein were relatively poorly expressed in patient tumors (Appendix A). We observed that BIM repression was relieved in the drug-treated U87 and GSCs (GSC12 Figure 2A; U87 Appendix A), suggesting that it may play a role in limiting apoptosis in GSCs. To test this, we used BIM siRNA combined with drug treatment and found that GSC12 displayed BIM-dependent apoptosis following MEKi+HDACi (Figure 2B). This demonstrates that repression of the pro-apoptotic BH3-only protein BIM plays a role in limiting cell death in these cells.

### 2.3. MEKi+HDACi Disrupts the Cell Cycle in GSCs

Cell cycle dysregulation is a well-established driver of apoptosis in cancer cells and HDACi disrupts the cell cycle in cancer cells specifically [44,45]. However, it is not known how the combined treatment of HDACi+RAS pathway inhibition affects the GBM cell cycle. To test if MEKi influenced HDACi-mediated changes to the GBM cell cycle (and its potential contribution to apoptosis), we analyzed our RNAseq data from GSC12 and U87 cells 24 h after treatment with MEKi, HDACi, or the combination of the two drugs. These treatments strongly affected the cell cycle factor gene expression in both U87 and GSC12, although the effects were different in the two cell lines (Appendix A). Of note, most of the observed cell cycle gene expression changes in GSCs were caused by HDACi, suggesting a modest contribution of MEKi.

To determine the effects of single and dual treatment on the progression of GBM through the cell cycle, we assessed the DNA content by propidium iodide staining and flow cytometry. MEKi+HDACi-treated GSCs displayed an altered DNA content, including an elevated sub-G1 population and an increased number of cells with the highest DNA content (>G2/M) (Appendix A). MEKi in U87 and LN229 cells displayed a suppressed S phase, while HDACi resulted in an accumulation of cells in G2/M (Appendix A). The analysis of cell cycle proteins pRB, cyclin B1, and pCDC2, which are indicative of specific cell cycle phases, confirmed that, compared to controls, the dual combination treatments resulted in marked changes to the cell cycle in the cultured cells (Appendix A). In U87 and LN229, the effect of combining MEKi with a selective HDAC1 inhibitor, mocetinostat (MOC), appeared to have a more acute effect than combining it with vorinostat on some cell cycle proteins (Appendix A). In both GSC12 and U87 cells, MEKi+HDACi disturbed the expression of critical mitotic genes (Appendix A). Overall, we interpret the altered mitotic DNA content and the simultaneous counterposing of upregulated and downregulated mitotic cell cycle factors as a serious disruption to the cell cycle, potentially contributing to apoptosis [46].

We considered the possibility that the cell cycle disruption may have been caused by apoptosis (rather than being a driver of apoptosis). To test this, we blocked the cleavage and activation of caspases using Q-VD-OPh in GSCs and assayed the levels of cyclin B1 and pRB. As expected, both CC3 and pH2AX were rescued by this treatment (Appendix A), again indicating that the DNA damage appeared subsequent to activation of the caspases. In contrast, we observed that the altered cyclin B1 and pRB were preserved upon Q-VD-OPh treatment with MEKi+HDACi (Appendix A), indicating that these changes do not rely on caspase activation and suggesting that cell cycle dysregulation may contribute to apoptosis.

### 2.4. Addition of a CD44 Ligand Partially Represses BMF and BIM and Reverses Apoptosis in GSCs

EGFR signaling is critical for most GBM’s growth and survival. We considered that MEKi+HDACi may be interfering with this signaling pathway. This was supported by the observation that HDACi resulted in a moderate reduction in EGFR gene expression (Appendix A). We therefore tested the possibility that providing additional growth and survival signals could reverse apoptosis induced by MEKi+HDACi. To broadly test this, we initially co-treated GSCs with 5% serum and MEKi+HDACi for 24 h. Adding serum resulted in a partial reversal of apoptosis, as indicated by reduced CC3 levels (Appendix A). Surprisingly, the addition of serum was associated with a reduction in the H3K9ac levels (Appendix A). These data suggested that a serum component may be able to revert HDAC inhibition and apoptosis.

We considered the possibility that a specific growth factor in serum may be responsible for this partial reversal of apoptosis. As all known growth factors in serum are smaller than 100 kDa, we used a molecular weight cutoff membrane to test if any of these were responsible for reversing the drug-induced apoptosis. Surprisingly, serum depleted of low molecular weight factors was still able to reverse apoptosis and reduce the H3K9ac levels, although it did not obviously impact the pERK levels (Appendix A). We concluded that serum factors reversed apoptosis through a MEKi- and growth factor-independent mechanism.

The prior literature suggested that hyaluronan (HA), the main glycosaminoglycan in the extracellular matrix, which is highly abundant in high molecular weight (HMW) fractions of serum, promotes the progression of glioblastoma (GBM) and cancer stem cell resistance to therapeutics [38,39,40]. We also observed that HDACi strongly repressed the main hyaluronic synthase, HAS2, in GSCs (Figure 3A). We therefore co-treated GSCs with HMW HA and MEKi+HDACi to test whether they reduced the cleavage of caspase 3. Remarkably, HMW HA partially reduced the CC3 levels (Figure 3B,C). The reduction was commensurate with that seen in serum+MEKi+HDACi-treated GSCs. We tested if HA might reverse the expression changes seen in cell cycle factors after MEKi+HDACi treatment. The addition of HMW HA did not have consistent effects on most of the cell cycle markers that we examined, suggesting that these changes were not associated with the HA-mediated reduction in apoptosis (Appendix A). We did note that HMW HA repressed the BH3-only pro-apoptotic protein BMF, and to a lesser extent BIM (Figure 3B,C). The decrease in BMF suggested the possibility that it may play a role in the increased survival of HA-treated GSCs.

### 2.5. Substituting BCL-2 Family Inhibitors for MEKi Induces Apoptosis in GSCs

Our data suggested that interactions between pro-apoptotic BH3-only proteins (e.g., BMF, BIM) with anti-apoptotic BCL-2 family proteins (BCL-XL, MCL1) regulate GBM cell survival. In particular, the changes in the BMF and BIM expression after HA and MEKi treatment suggested that BCL-XL (and possibly MCL1 and BCL-w, Figure 2A) may be inhibiting apoptosis. To test this, we used navitoclax (NX), a BH3 peptide mimetic that antagonizes BCL-XL, BCL-2, and BCL-w at nanomolar levels. We carried out this experiment with or without MEKi and/or HDACi. These experiments revealed a significant increase in CC3 levels specifically when HDACi was combined with NX (Figure 4A). These data suggest that the inhibition of BCL-2 family proteins may enhance the efficacy of HDACi in GBM.

Our RNAseq analysis also indicated that, in GBM cells, the most highly expressed pro-survival gene of the BCL-2 family was MCL1 (Figure 2A). Therefore, to test if MCL1 played a key role in the survival of GSCs, we used a selective MCL1 inhibitor, S63845, alone or in combination with HDACi. These experiments revealed that HDACi could significantly enhance the levels of CC3 induced by MCL1 inhibition (Figure 4B). An analysis by annexin V showed elevated levels of apoptosis in the HDACi+MCL1i-treated GSC39 cells, confirming apoptosis as a major cell death pathway induced by the drug treatment (Figure 4C). These data indicate that the ability of MCL1 to promote GBM survival is markedly compromised by inhibiting HDACs.

## 3. Discussion

Glioblastomas invariably relapse, usually less than 16 months after treatment [47]. Finding approaches to improve these outcomes is a high priority. We found that GBM cells were sensitive to HDACi when combined with MEKi or BCL-2 inhibition (Figure 4D). The HDAC inhibitor vorinostat inhibits GBM growth in combination with other treatments and has been tested in clinical trials [48,49].

The concentrations of vorinostat used in our study are similar to those used for in vitro studies which lead to FDA approval for lymphoma [50]. Interestingly, vorinostat was recently identified in a study of diffuse intrinsic pontine glioma (DIPG) as having a relatively high CNS multiparameter optimization desirability score [51]. This is consistent with prior reports of its moderate blood–brain barrier penetration [52,53,54] and supports previous reports that HDAC inhibition can synergize with MEKi as a potential therapeutic strategy in neurological cancers [51].

Several HDAC inhibitors (vorinostat, panobinostat, romidepsin, belinostat) are FDA approved for various indications [55]. Although the efficacy of broad spectrum HDACi in GBM may be limited by the potential for reactivation of the DNA repair enzyme MGMT [56], recent studies have indicated that selective HDACi may improve the therapeutic window [57]. In this regard, the selective inhibition of HDAC1, which is essential for glioma stem cell survival [58], was more effective than vorinostat in dysregulating cell cycle proteins, and several HDAC inhibitors displayed combinatorial effects with MEKi in promoting apoptosis.

Histone deacetylase inhibitors (HDACi) promote caspase-dependent apoptosis by diverse mechanisms [59,60,61]. As with prior work, our study of MEKi combined with HDACi highlighted the reliance of GBM cells on BCL2 proteins [62,63,64,65,66,67]. One of the most striking results in our study was the marked increase in cell death in the presence of HDACi plus an MCL1 inhibitor. To our knowledge, our study is the first to show that combining MCL1 inhibition with HDACi in GBM causes significant apoptosis in glioma stem cells. Our observations are made more relevant by the recent demonstration that GSCs are particularly reliant on MCL1 expression and highly sensitive to its inhibition [67]. Importantly, BCL-XL and MCL1 inhibitors have progressed to Phase II clinical trials [68] (CL1-64315-004-2019-004896-38) but have not been tested with HDACi in GBM. An important remaining question is whether the cell cycle defects observed in these cancer cells from HDACi are responsible for potentiating the effects of BCL-2 family inhibitors. In this regard, normal cells are able to repair or cope with the effects of these epigenetic inhibitors [44,45], and future studies that directly compare them with GSCs would be informative.

HDAC inhibition in vivo combined effectively with MEKi, leading to a long-term decrease in melanoma and colorectal tumor growth [69,70]. Our data demonstrate that MEKi+HDACi significantly disrupts glioma cell cycle factors. Cell death resulting from cell cycle dysregulation remains incompletely understood. Our data on GBM suggest that the expression changes in critical cell cycle factors occurred prior to the cleavage of the caspases (Appendix A), suggesting that cell cycle defects may trigger apoptosis independently of DNA damage. Our study did not define the exact mechanism by which this may be occurring. However, there appeared to be counterposing effects of the drug combination on cycling: MEKi impaired G1/S phase progression, while HDACi appeared to impact exit from the G2/M phase. An immunoblot analysis indicated that p21 was upregulated, which is predicted to inhibit G1 transition; yet, at the same time, the pRB levels were significantly diminished, which is associated with progression of the cell cycle. These contradictory mitogenic and cell cycle arrest signals may create unresolvable conflicts, contributing to apoptosis [46]. Additionally, since BIM is normally degraded during mitosis [71], it is possible that, when BIM or BMF expression is rescued by MEKi during an impaired G2/M phase caused by HDACi, this could trigger apoptosis. One limitation of our observations is that some of our analyses were determined on a population of cells. We were not able to distinguish differing effects in those single cells, and future studies of this may be informative.

Cell culture-adapted GBMs have been studied for their survival response to the dual treatment of trametinib and entinostat [72] using a cell-permeable dye to identify dead cells. The mentioned study suggested that the combination may be more effective than single treatments alone at promoting apoptosis. Our study extends those initial findings to a potentially more clinically relevant model by using multiple recently derived patient GBM isolates grown under glioma stem cell conditions. We further identify apoptosis as a major cause of cell death and also reveal pro-survival dynamics regulating these responses.

The ability of HMW hyaluronan to repress BMF and reverse apoptosis induced by MEKi+HDACi was surprising. This reversal was not associated with rescuing HDAC activity or MEK1/2 signaling, nor was it consistently associated with the rescue of cell cycle factors. Intriguingly, the HA treatment was accompanied by decreased levels of BMF and BIM, suggesting that their combined reductions may promote this increased survival. HA production by glioma is associated with proliferation, therapy resistance, and elevated tissue invasion [39]. HA has been shown to impact cellular adhesion [73] and an unexplored aspect of this is whether HA-GSC interactions limit anoikis, the cell death program associated with the loss of attachment to an extracellular matrix. Thus, in addition to its effect on BMF levels, HA may also affect the sequestration of BIM and BMF to the cytoskeleton [74], impacting anoikis, and potentially mitigating the efficacy of BIM and BMF to promote apoptosis [75,76]. Future studies of BIM and BMF interactions with microtubules and actin in HA-treated GSCs may be informative about additional mechanisms of apoptotic repression.

An important question is whether there is a connection between the reversal of apoptosis by HA to the cell cycle defects induced by HDACi. One possibility stems from the fact that the HA receptor RHAMM has been shown to regulate mitotic spindle integrity [77]. Thus, if signaling from RHAMM is elevated by the addition of HMW HA in these nonadherent GSCs, it may have the potential to mitigate some aspects of defective mitosis, leading to a partial reversal of apoptosis.

Overall, our study warrants more in-depth tests of the efficacy and mechanisms of action of combining MEKi+HDACi and HDACi+BCL-2 family inhibitors in GBM, possibly by employing local drug delivery or intrathecal chemotherapy to circumvent the blood–brain barrier [67,78,79,80]. Our study also suggests that further research on glioma stem cell-HA interactions may reveal important pro-survival dynamics which regulate GBM.

## 4. Materials and Methods

### 4.1. Cell Lines and Cell Cultures

GSC cells were obtained from the Mayo Clinic Brain Tumor Patient-Derived Xenograft National Resource [81] and grown in stem-like media containing Neurobasal or DMEM/F12 media (Gibco #21103049), with 10 mL B27 without Vitamin A (Gibco #12587010), MEM NEAA (Gibco #11140-050), 10 μg EGF (Gibco #PHG0314), 10 μg FGF (Gibco #13256-029), 1 mM sodium pyruvate (Gibco #11360-070), 2 mM GlutaMAX-I (Gibco #3505-061), 100 units/mL penicillin, and 100 mg/mL streptomycin (Gibco #15140122) passaged using glioma stem cell/brain tumor-initiating conditions [82,83] where neurospheres in suspension were collected and centrifugated at 200× *g* for 5 min before removing the media and resuspending pellet in 1 mL or less 0.25% Trypsin, 2.21 mM EDTA, without sodium bicarbonate (Corning #25-053-Cl) for 3–5 min incubated at RT. Trypsin was then diluted with at least 10x volumes of Neurobasal-only medium. Cells were centrifuged and washed once more with Neurobasal medium, spun again, then resuspended in complete GSC media. GBM cell culture adapted lines U87 and LN229 were obtained from American Type Culture Collection and were cultured in DMEM High Glucose (GE Life Sciences #SH30081.02) with 2 mM L-Glutamine (Corning #25-005-Cl), 10% Fetal Bovine Serum (Corning #35-010-CV), 2 mM GlutaMAX-I (Gibco #3505-061), 100 units/mL penicillin, 100 mg/mL streptomycin (Gibco #15140122), and 1 mM sodium pyruvate (Gibco #11360-070).

Inhibitors used were from APExBIO: trametinib (#A3887), vorinostat (#A4084), Z-VAD-FMK (#A1902), Q-VD-OPh (#A1901), entinostat (#A8171), mocetinostat (#A4089), and LMK235 (#AA4494). High molecular weight hyaluronan purchased from Bio-Techne/R&D system (GLR002) was resuspended in GSC media at 10 mg/mL. GSCs were treated for 24 h with 5 mg in a total of 2.5 mL media at the time of drug addition.

### 4.2. Cell Lysis and Immunoblots

After removing media from cells in a 6-well plate, 1.5 mL of ice-cold PBS was added, and the cells were scraped into a 1.7 mL tube. Cells were collected by centrifugation at 500× *g* for 4 min, PBS was removed, and cells were placed on ice for lysis, or snap-frozen in LN2 and stored at −80 °C. Cell pellets were lysed in 10 mM Tris-Cl (pH 8.0), 150 mM sodium chloride, 1% Triton X-100, 1 mM EDTA, and 3 μL per 100 μL Complete Protease Inhibitor (Sigma #P8340). Samples were incubated on ice for 20 min then centrifuged for 20 min at 13,000× *g*. Protein concentration was measured using DC™ Protein Assay Kit (Bio Rad # 5000116). Absorbance readings at 750 nm using a BioTek Synergy 2 plate reader were used to calculate mg/mL protein values. Samples were made up of NuPage LDS sample buffer (Invitrogen, NP0007) and Invitrogen Novex 10 X Bolt Sample Reducing Agent (Thermo Fisher Scientific #B0009), and were incubated at 95 °C for 7 min before loading equal protein amounts onto Mini-PROTEAN TGX Stain-Free Gels 4–20% Tris-Glycine polyacrylamide gels (Bio Rad #4568096). Gels were run in TGS buffer for 35 min at 100 volts and transferred to TransBlot^®^ Turbo™ Mini-size nitrocellulose membranes (Bio Rad #1704158) for 5 min using the Trans-Blot^®^ Turbo™ RTA Transfer Kit, Nitrocellulose System (Bio Rad #170-4270), and Trans-Blot Turbo Transfer Buffer (Bio Rad #10026938). Membranes were blocked in 5% non-fat dry milk or in StartingBlock™ (TBS) Blocking Buffer (Fisher #37542) with orbital shaking for 30 min at room temperature. Antibodies were incubated with blots overnight at 4 °C with orbital shaking, followed by antibody removal and washing in 10 mL of TBS-T three times for 5 min at RT with orbital shaking. Primary antibodies were detected by addition of species-specific horseradish-peroxidase (HRP)-conjugated secondary antibody in blocking buffer, and incubated with orbital shaking for 60–120 min followed by washing as for primary antibodies. After removing the last wash buffer, chemiluminescent visualization (SuperSignal™ West Pico PLUS Chemiluminescent Substrate, Thermo Fisher Scientific Catalog #34578) was used. Membranes were visualized on an Invitrogen iBright 1500. Membranes could then be stripped after rinsing the membranes three times with DI water using a low-pH stripping buffer (25 mM Glycine, 1% SDS, pH 2.3) twice for 15 min on high agitation. Membranes were then rinsed twice in 5mL TBS-T and washed in 10mL TBS-T for 5 min on the orbital shaker before the new primary antibody was added. Antibodies used for immunoblots were from Cell Signaling Technologies: ERK1/2 p44/42 MAPK (Erk1/2) (#9102S), phospho-ERK1/2 (pERK; Phospho-p44/42 MAPK (Erk1/2) (Thr202/Tyr204) (#4370S), BIM (#2933), Phospho-Histone H2A.X (pH2AX; (Ser139) (#9718), Cleaved Caspase-3 (CC3, #9664), Acetyl-Histone H3 (H3K9ac; Acetyl-Histone H3 (Lys9/Lys14) Antibody #9677), Acetyl-Histone H3 (H3K27ac; Acetyl-Histone H3 (Lys27) #8173), BCL2 (#4223), MCL1 (#94296), Bcl-xL (#2764), pRB (#9308), Cyclin B1 (#12231), pCDC2 (#4539), and p21 (#2947). Dilutions for all primary antibodies were 1:1000 with a 1:2000 dilution for secondary antibodies. For housekeeping genes, dilutions for primary β-Actin HRP Antibody (sc-47778) were used at 1:200 and GAPDH Cell Signaling Technologies (#5174S) at 1:1000.

### 4.3. Gene Knockdown Using Small Interfering RNA (siRNA)

BIM ON-TARGETplus Human BCL2L11 siRNA–SMARTpool, 5 nmol (L-004383-00-0005) was purchased from Dharmacon/Horizon Discovery. GSCs were trypsinized immediately prior to siRNA transfection with 50 pmol BIM siRNA or non-targeting siRNA (D-001810-10-05) in a total of 0.5 mL of Optimem (Thermo Fisher, 11058021) with 12 μL RNAiMAX (Thermo Fisher 13778150) and added to cells in 2 mL of GSC media. The following day, media was changed. After 48 h, cells were treated with MEKi or HDACi for 24 h followed by analyzing BIM knockdown and cleaved caspase 3.

### 4.4. Growth and Cytotoxicity Curves

For growth curves, U87 cells were seeded at 10,000 or 5000 cells/well in a 96-well plate and treated with vehicle trametinib, vorinostat, or both after 12 h. Cells were subsequently imaged at a 10x magnification every 12 h for 6.5 days using the BioTek Cytation 5 automated cell counter equipped with a BioSpa 8 incubator. Cell nuclei were counted from a 200 µm offset image using BioTek Gen5 software to generate cell counts. For cytotoxicity curves, U87 cells were seeded at 2500 cells/well in a 96-well white plate with transparent bottom and, after 12 h, dosed with trametinib, vorinostat, or both utilizing the same dosages as the growth curves. Followed by a 6-day incubation, CellTiterGlo 2.0 (Promega #G241) was used to measure cell viability by using the BioTek Synergy 2 plate reader for luminescence. IC50 values were calculated using GraphPad Prism 9. Luminescence values were organized in a matrix and uploaded into SynergyFinder in order to visualize synergy data. CI values were calculated using CompuSyn where, first, the raw luminescence values were normalized to the luminescence of no-drug control before being input as the effect into the program. The drug combo ratio used was 1:4266.67 to take into account the highest concentration of trametinib (15 nM) compared to that of vorinostat (64 μM).

### 4.5. Annexin V Flow Cytometry

U87 and LN229 cells were seeded 1 × 10^6^ cells in a 10 cm dish. After 24 h of drug treatment, media was removed and placed in a labeled 15 mL falcon tube. U87 and LN229 were collected through tryspin-EDTA dissociation and washed twice with PBS without calcium or magnesium. Cell numbers were counted on a BioRad Automated Cell Counter or Countess cell counter (Thermo Fisher). Cells were pelleted at 500× *g* (or 200× *g* for GSCs) for 10 min. The cell pellet was resuspended in Annexin V binding buffer (BD Pharmingen, 556454) and 5 × 10^5^ cells were transferred to a FACS tube containing 5 µL propidium iodide (PI) staining solution (Peprotech 60910-00) and 5 µL FITC-Annexin V (BD Pharmigen, 556419). Cells were incubated for 15 min in the dark. After incubation, 400 µL of Annexin V Binding buffer was added and FACS was collected on a BD LSRII or LSR Fortessa and analyzed using FlowJo v10. Compensation controls used for this analysis included unstained cells, and cells stained for each individual fluorophore (Cy5-PI and FITC-Annexin V). Gating strategy eliminated cell debris and doublets through forward and side scatter plots.

### 4.6. Cell Cycle Analysis by Flow Cytometry

U87 and LN229 cells were seeded in a 6-well plate and incubated with vorinostat, trametinib, or a combination of both for 24 h. Following this, the media was aspirated and cells were dissociated using trypsin-EDTA. The cells were washed twice with ice-cold 1X PBS. Cell numbers were counted on a Countess (Thermo Fisher) and were pelleted at 200× *g* for 10 min. Cells were resuspended at 2 × 10^6^/1 mL of ice-cold PBS and fixed in 9 mL of 70% ethanol. The cells were stored at −20° Celsius for 24 h. The cells were then pelleted at 500× *g* for 10 min and washed with 1X PBS. The cells were stained with 400 μL of staining solution composed of 0.1% *v*/*v* Triton-X 100 in PBS, 2 mg DNAse-free RNase A (Thermo Fisher EN0531), and 500 μg/mL propidium iodide (Biogems 60910-00), and incubated for 15 min at room temperature. The DNA content was measured on BD LSRII or LSR Fortessa and analyzed using FlowJo v10. Gating on FSC-A vs. SSC-A plot, FSC-H vs. FSC-W plot, and SSC-H vs. SSC-W plot was used to eliminate debris and any doublets from the data.

### 4.7. RNAseq Analysis

Library preparation was performed on purified, extracted RNA using a KAPA mRNA HyperPrep Kit (Kapa, Biosystems, Wilmington, MA, USA) according to the manufacturer’s protocol. Two randomized 12-sample pools at a concentration of 10 nM were created for sequencing. These pools were diluted to 1.65 pM and spiked with 1% PhiX bacterial genome as a positive control for alignment. High throughput sequencing with 75-bp single-end reads was performed on an Illumina NextSeq 550 using an Illumina NextSeq 500/550 High Output Kit. Reads were aligned to the human transcriptome GENCODE v35 (GRCh38.p13) using STAR and counted using Salmon [84,85]. Normalization and differential expression analyses were performed using the R package DESeq23 [86]. Genes where there were fewer than three samples with normalized counts less than or equal to five were filtered out of the final data set. Conditions were run in biologic triplicate.

## Figures and Tables

**Figure 1 ijms-24-13688-f001:**
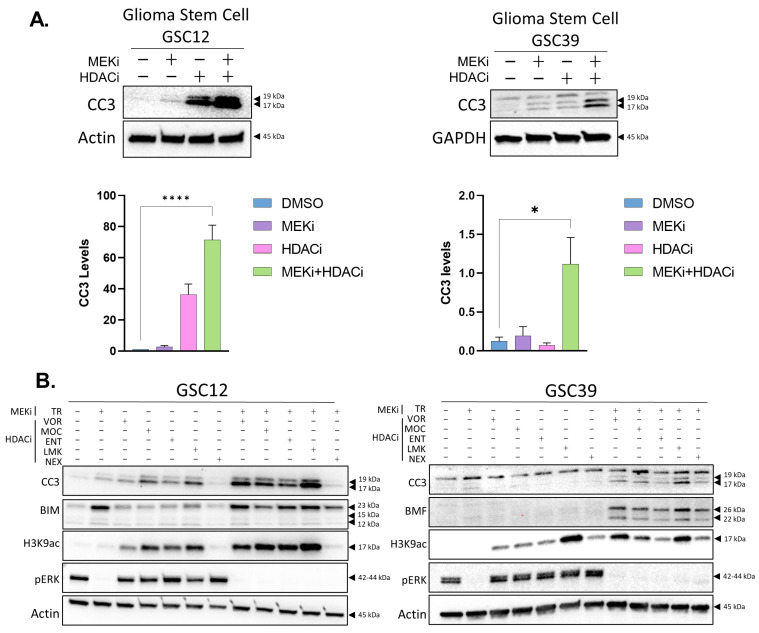
Combining MEKi and HDACi induces apoptosis in glioma stem cells. (**A**) Western blot for apoptotic marker cleaved caspase-3 (CC3) in GSC12 and GSC39 cells. Cells were treated for 24 h. GSCs were treated with HDACi (2.5 μM vorinostat) and MEKi (25 nM trametinib); β-Actin or GAPDH protein was used as a loading control and for normalization. The graphs quantify CC3 expression levels. Graphs portray the mean and SEMs for the percentage of CC3 levels (n = 3). * *p* = ≤ 0.05; **** *p* ≤ 0.0001, Student’s *t*-test. (**B**) To determine the contribution of specific HDACs to GBM cell survival, GSCs were treated with selective HDAC inhibitors: mocetinostat (MOC, a HDAC 1/2/3 inhibitor, 1 μM), entinostat (ENT, a HDAC 1/3 inhibitor, 2 μM), LMK235 (LMK, a HDAC 4/5 inhibitor, 1 μM), and nexturastat (NEX, a HDAC 6 inhibitor, 2 μM); cells were treated for 24 h with or without MEKi.

**Figure 2 ijms-24-13688-f002:**
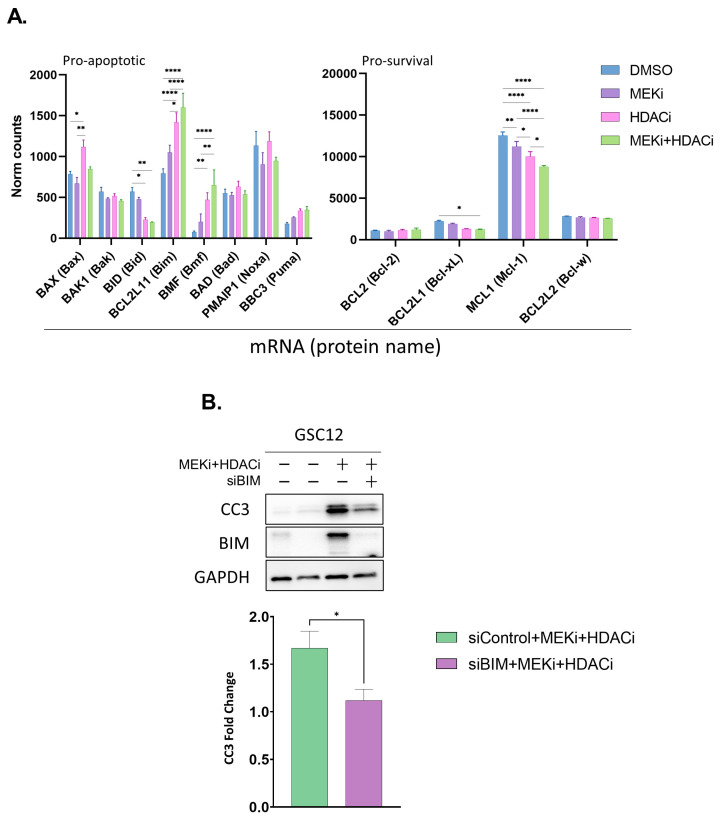
MEKi+HDACi significantly alters expression of major pro- and anti-apoptotic factors in GSCs. (**A**) RNA-sequencing data for pro-apoptotic (**left**) and anti-apoptotic (**right**) genes in GSC12 glioblastoma cells following 24 h incubation with 25 nM TR and/or 2.5 μM VOR. Data are means + SEM (n = 3). Significance was tested using a multiple comparison one-way ANOVA * *p* ≤ 0.05; ** *p* ≤ 0.01; **** *p* ≤ 0.0001. (**B**) Western blots for CC3 and BIM in GSC12 cells treated with 25 nM TR and 2.5 μM VOR for 24 h following a 72 h transfection with BIM siRNA. The graph quantifies CC3 expression levels only for the drug+siRNA treated cells. Data are means + SEM (n = 3). * *p* = ≤ 0.05, *t*-test.

**Figure 3 ijms-24-13688-f003:**
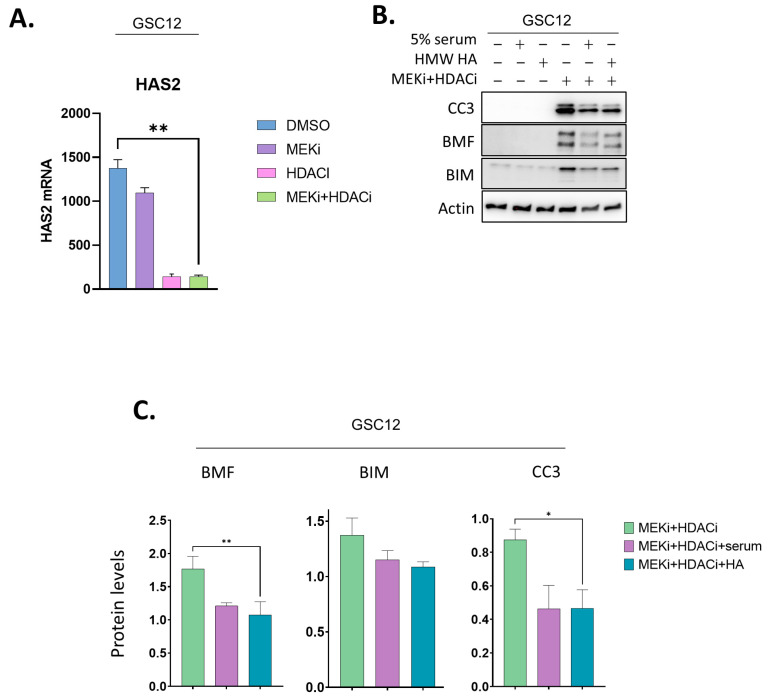
Rescue of apoptosis by high molecular weight hyaluronan coincides with repression of pro-apoptotic BMF. (**A**) RNAseq values for HAS2 mRNA expression in GSC12 cells +/− MEKi and HDACi (25 nM trametinib, 2.5 μM vorinostat). (**B**) GSC12 cells were treated with MEKi+HDACi with or without 5% serum or HMW hyaluronan followed by immunoblot for CC3, BMF, BIM, and actin. (**C**) Graphs quantify the levels of CC3, BMF, and BIM in three independent experiments of (**B**). Data are means + SEM. * *p* < 0.05; ** *p* ≤ 0.01 *t*-test.

**Figure 4 ijms-24-13688-f004:**
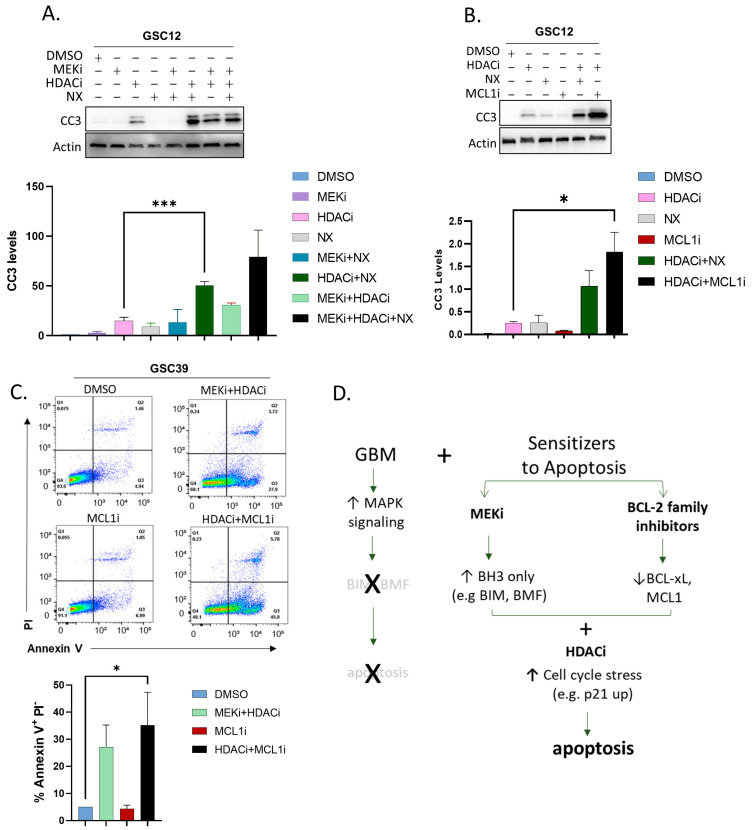
Replacing MEKi with BCL-2 family inhibitors triggers apoptosis. (**A**) GSC12 was treated for 24 h with the indicated combinations: MEKi (25 nM trametinib), HDACi (2.5 μM vorinostat), NX (100 nM navitoclax) followed by analysis of apoptosis (CC3). (**B**) GSC12 was treated for 24 h with the indicated combinations: MEKi (25 nM trametinib), HDACi (2.5 μM vorinostat), NX (100 nM navitoclax), MCL1i (100 nM S63845) followed by analysis of apoptosis (CC3). Actin protein was used as a loading control for (**A**–**C**). GSC39 cells were treated for 24 h with the indicated combinations: MEKi (25 nM trametinib), HDACi (2.5 mM vorinostat), MCL1i (100 nM S63845) followed by analysis for apoptosis by annexin V-propidium iodide (PI) flow cytometry measuring apoptosis (annexin V positive/PI negative, Q3). Graph depicts mean +/− SEM, (n = 7). (**D**) Diagram summarizing results highlighting the use of HDACi to interfere with GBM cell cycle combined with potentiating intrinsic apoptosis using either MEKi to rescue BH3-only proteins, or inhibitors of pro-survival BCL-2 family proteins. Graphs portray means + SEM. *p* = * ≤ 0.05; *** ≤ 0.001, *t*-test.

## Data Availability

Data are available upon request.

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
