# Peer review of "Glioma Stem Cells Are Sensitized to BCL-2 Family Inhibition by Compromising Histone Deacetylases"

_ijms, 2023, doi:10.3390/ijms241813688_

Round 1

Reviewer 1 Report

In this study, the authors examined the effects of the combination of HDAC inhibition and MEK1/2 inhibition or BCL-2 family inhibition on apoptosis of glioma stem cells and cell lines. The combination of MEKi and HDACi suppressed the cell growth, induced cell cycle defects and apoptosis, and rescued the expression of BIM and BMF. Furthermore, the combination of MCL1 inhibition with HDACi induced significant apoptosis in glioma stem cells. The experiments were well-done, and the study may provide useful information for the development of treatment. The following comments need to pay attention.

1. There is a mix of cells in the supplemental data in the legends of some figures. This is confusing to the readers and should be corrected. For instance, Figure 1 does not contain the results of U87, which is in Figure S2. On the other hand, Figures S2 and S5 do not contain the results of GSC12 though stated in the legends of Figures S2 and S5.

2.  Some graph legends are missing p-values, but include them where appropriate. Also, clearly state what the error bars in the data for all graphs indicate.

3.  In Figure 1B, BMF was analyzed using GSC39, but this description is missing in the results section of page 2, line 87-88.

4. Although the analysis of TK1 and cyclin E1 was described in page 6, line 164, the blots of TK1 and cyclin E1 are missing in Figure S9B.

5.  The labels of Y-axes are missing in the graphs of Figures 3C and S10B.

6. Though stated in the legend, the blot of pENSA is missing in Figure S10D.

7.  In Materials and Methods: Anti-GAPDH antibody is missing. There is no description of the statistical processing method with the software. They should be included.

Author Response

We are grateful for the reviewer's thoughtful reading of our manuscript and apologize for the oversights. Below we describe how we have addressed each point that has been raised and have highlighted changes in the manuscript.

Reviewer #1:

  1. We have corrected the mislabeling of cell lines in the figure legends as pointed out by the reviewer.
  2. We have ensured all graph legends have p-values and statements regarding error bars.
  3. We have added a description to the main text at line 80 for the BMF result for GSC39.
  4. We have removed the text referring to the antibodies that we did not test.
  5. We have corrected the missing axes labels for Fig 3C and S10B.
  6. We have removed the description of pENSA.
  7. We have added a description to the methods to include the source of GAPDH antibody.

Reviewer 2 Report

The study by Merati et al investigates the combination of HDAC inhibitors with MEK inhibition and bcl-2 family inhibition in vitro for the treatment of glioblastoma.

The study is well written and experiments appear to have been thoroughly performed. Although combinations of HDACi with both MEKi and Bcl-2 family inhibitors have been previously evaluated, the current study provides some interesting observations.

Major concern: While the experiments in the first half of the paper (Figures 1 and 2 plus Supplementary Figures) were performed in several cell models (there seems to be a mix up, however, of what is shown in the main Figures and what is shown in the supplementary figures compared to the figure legends), the experiments in the second half of the paper (Figures 3 and 4) seem to have only been performed in one cell model (GSC12). Confirmatory experiments in a least one more cell model would be important to be able to generalize the results.

Concerns to be addressed:

The legend to Figure 1 mentions U87 cells but I could not find these data in the figure panels

In Figure 2 A it is not clear whether the data shown are from U87 or from GSC12 or are averaged across both cell lines. Lines 142 and 143 seem to state the same thing.

Figure 4: Although “GSCs” is used in the legend, the data shown seem to be only from GSC12

Line 136: typo in demonstrates

Line 360/361: 10 g EGF, 10 g FGF cannot be the correct units.

Author Response

We are grateful for the reviewer’s thoughtful reading of our manuscript and apologize for the oversights. Below we describe how we have addressed each point that has been raised. We have highlighted the changes in the manuscript.

Reviewer #2:

We thank the reviewer for pointing out the missing labelling for GSC39 in Fig 4C. We hope the reviewer will agree that together with GSC12 in Fig 4A-B, our study supports the conclusion that HDACi combined with inhibition of MCL-1 induces apoptosis in GSCs.

  1. We apologize for the confusing labelling in the Fig 1 legend and have corrected the legend. The U87 data are in Fig S2.
  2. We have corrected the description in the Figure legend to clarify that the data are from GSC12.
  3. We have clarified that the data in Fig 4A-B are from GSC12 and while Fig 4C is from GSC39.
  4. We have corrected the typographical error in the word demonstrates on line 136.
  5. We have corrected the concentration of EGF and FGF in the methods.

Round 2

Reviewer 1 Report

The authors have revised the manuscript according to the reviewer’s comments appropriately. There is no criticism.

Reviewer 2 Report

concerns have been sufficiently addressed.